# Rice Husk Ash as Pore Former and Reinforcement on the Porosity, Microstructure, and Tensile Strength of Aluminum MMC Fabricated via the Powder Metallurgy Method

Azmah Hanim Mohamed Ariff [1,2,]*, Ong Jun Lin [1], Dong-Won Jung [3,]*, Suraya Mohd Tahir [1] and Mohd Hafis Sulaiman [1]

1 Department of Mechanical and Manufacturing Engineering, Faculty of Engineering, Universiti Putra Malaysia, Serdang 43400, Selangor, Malaysia
2 Research Center Advance Engineering Materials and Composites (AEMC), Faculty of Engineering, Universiti Putra Malaysia, Serdang 43400, Selangor, Malaysia
3 Department of Mechanical Engineering, Jeju National University, 1 Ara 1-dong, Jeju 690-756, Korea
* Correspondence: azmah@upm.edu.my (A.H.M.A.); jungdw77@naver.com (D.-W.J.)

**Abstract:** The handling of rice husk ash (RHA) has been raising environmental concerns, which led to the consideration of incorporating RHA in aluminum metal matrix composite fabrication. Due to the high silicon dioxide content of RHA, it can assist in enhancing both the properties and functionality of pure aluminum. In this research, the fabrication of aluminum metal matrix composite was carried out by utilizing different compositions of RHA, including weight fractions of 10 wt.%, 15 wt.%, and 20 wt.% via a powder metallurgy approach. The element powders, including aluminum and RHA, and magnesium stearate as a binder, were mixed, compacted, and sintered to attain a composite sample in the form of a pellet. The pellet was then characterized using field emission scanning electron microscopy (FESEM-EDX) to identify the pore structure and size for each RHA composition. The samples were also mechanically tested via Archimedes' Principle and Brazilian Testing to identify their density, porosity, and tensile strength, respectively. The total porosity of RHA-15 wt.% was found to be the highest at 19.19%, yet with the highest tensile strength at 5.19 MPa due to its low open porosity at 4.65%. In contrast, the total porosity of RHA-20 wt.% was found to be slightly lower at 15.38%, with the highest open porosity at 6.95%, which reduced its tensile strength to 5.10 MPa, therefore indicating that reducing open porosity through controlling the composition of reinforcement tends to enhance the mechanical strength of aluminum metal matrix composites.

**Keywords:** porous aluminum; aluminum metal matrix composite; rice husk ash; powder metallurgy; microstructure; mechanical properties; porosity

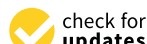



## 1. Introduction

Rice Husk Ash (RHA) had been gaining attraction due to its value in various applications from the automotive to the manufacturing industry. Generally, the primary advantage of RHA is its tendency to improve the mechanical properties and characteristics of porous metal due to the presence of silicon dioxide within its chemical constituents. RHA and its superior properties have served in industries from gas/liquid filters, catalyst carriers, and removal of diesel particulates to lightweight structures [1]. RHA is produced from the burning of rice husk (RH) which is found at the outermost layer of paddy grains that were separated from milled rice grains during the milling process [2]. With the aid of a high-quality boiler, RHA can be produced effectively and thus serves a great advantage in not only energy generation through combusting RH but also can be utilized in silica production, as mentioned above. Moreover, depending on the technological capabilities as well as burning conditions, RHA tends to be presented in different colors varying from

greyish white to pink depending on its reactiveness and the presence of chemical constituents [1,3]. Abdullah et. al., in their study, discuss the effect of different temperatures and chemical treatments on the RHA formed [3]. Therefore, through consideration of the environmental advantage and superior properties of RHA, it was selected and utilized in the fabrication of porous aluminum in this study.

Porous metal is typically referred to as metal with voids within the metal matrix which allow the targeted metal element to achieve both lightweight and sturdy properties with promising strength and hardness as well as resistance to wear and tear. Similarly, porous aluminum provides an even more favorable weight due to its extravagantly low specific weight of merely 2.7 gm/cm$^3$. Nevertheless, the advantage also serves as a great drawback to the manufacturability of aluminum products, which thus leads to a great hurdle for manufacturing industries to fully cultivate the potential and usage of pure aluminum in manufacturing industries [4]. In recent years, various approaches have been explored in the manufacturing of porous aluminum with different fabrication methods and reinforcement materials which opens many possibilities for the manufacturing, construction, automotive, and building industries [5–7]. For instance, open-celled porous aluminum had been depicting great functionality in sound damping elements, heat exchanger elements, filters, and design elements, which are particularly adopted in the building and construction industry [5]. In contrast, closed-celled porous aluminum including aluminum-based alloys can be found widely in the manufacturing of structural and internal parts of vehicles such as submarines, aircraft, mid-air transportation, and automotives [8].

As mentioned, various fabrication methods had been adopted in ensuring the effective and efficient production of porous aluminum for either open-celled or closed-celled aluminum. One of the most cost-effective approaches to producing porous aluminum is through powder metallurgy. According to Kheradmand et al., powder metallurgy has various benefits, including a high fabrication rate, homogeneous microstructure fabrication, a reduced number of secondary steps, and a working range for the fabrication of large alloy composites which makes it extremely applicable to the fabrication of porous aluminum [9]. Cañadilla et al. [10] successfully produced porous aluminum with a tensile strength of 281.6 MPa and porosity of 60.8% using saccharose solution as a pore-forming agent at a volume percentage of 70 vol.%. As for Kumar and Golla [11], sodium chloride was utilized in the production of porous aluminum with a tensile strength of 1590 MPa and a porosity of 34.16%. Although the characteristics of porous aluminum may differ depending on the applied reinforcement material and process parameters, it was recorded that porosity and strength tend to vary as the composition of reinforcement material increases [12]. Various research has been carried out as well, particularly on the fabrication of porous aluminum with industrial waste, which includes rice husk ash and fly ash. Xavier et al. reported that the fabrication of aluminum matrix composite with economical and environmentally friendly materials such as rice husk ash and fly ash reveals that the wear behavior and mechanical properties were found to be improved as the reinforcement material composition increases [13]. Wear behavior and tribological studies were also carried out by Ghosh et al., who further verified that the presence of industrial waste in the form of nanoparticles allows the production of samples with optimum mechanical and tribological behavior, which, in the meantime, allows the proper handling of industrial waste, leading to the improvement of environmental issues and concerns [14].

Porous aluminum has been utilized in various industries and sectors. However, complex techniques and issues circulating costs greatly reduce the market value of porous aluminum in bulk production and fabrication. This leads to the consideration of adapting RHA as a pore-forming agent with consideration of both its economic and environmental value. Apart from that, though a large number of studies can be found revolving around the fabrication of aluminum matrix composites with RHA, minimal research was found that generally focuses on the fabrication of porous aluminum through a powder metallurgy approach [15]. Therefore, this study aims to produce porous aluminum with RHA as a pore-forming agent via the powder metallurgy method as well as to understand the effect

of its composition on produced metal matrix composites. RHA compositions of 10 wt.%, 15 wt.%, and 20 wt.% were used and added to pure aluminum during the fabrication process. The microstructure, density, total porosity, and tensile strength values of the respective samples were then observed and evaluated.

## 2. Materials and Methods

### 2.1. Sample Preparation

Aluminum, Al (purity: 99.9%, average particle size: 200 nm), and rice husk ash (RHA) were bought from a commercial source specifically from Nilai, Negeri Sembilan. Figure 1 shows the FESEM images of the raw materials. The received RHA was first processed and treated individually via a blending and sieving process to ensure uniformity of the particle sizes, which serves as a major factor in attaining the ideal and uniform porosity of the end product. The obtained RHA was ground to a smaller particle size using a mechanical blender or National MX-896TM dry blender to reduce the overall particle size before sieving with a 106 μm Retsch 200 × 50 mm 106 μm Test Sieve which gave rise to the production of rice husk ash with an average particle size of 106 μm. The chemical composition of RHA is presented in Section 3.

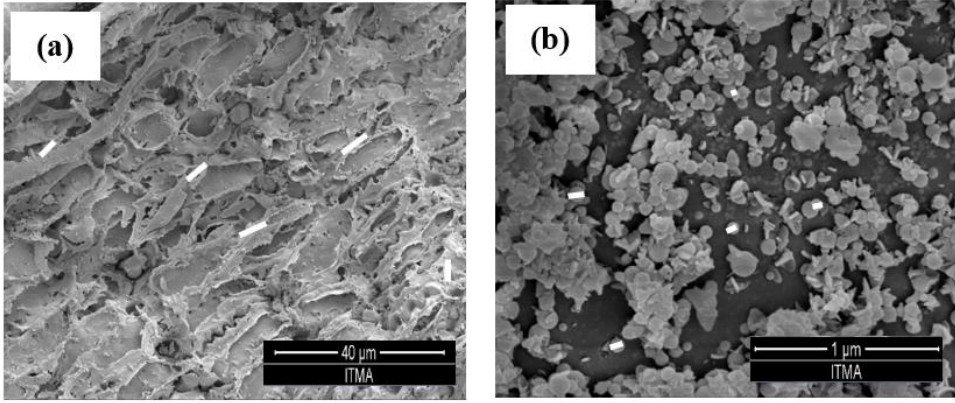

**Figure 1.** (**a**) RHA after grinding process, and (**b**) aluminum powder.

Three different samples were prepared with three different RHA compositions including RHA-10 wt.% and Al-85 wt.%, RHA-15 wt.% and Al-80 wt.%, and RHA-20 wt.% and Al-75 wt.%, as referred to by Shaikh et al. [8]. Each bulk sample was added to 5 wt.% magnesium stearate, which acts as the binder, before moving on to the ball-milling process. The prepared samples were homogeneously mixed using a planetary micro mill-pulverisette 6 (PM 100, Retsch, Haan, Germany) at a rotation speed of 400 rpm for 2 h with alternating directions (1 h clockwise and 1 h anticlockwise) in zirconium oxide (94.2% $ZrO_2$) grinding bowls and balls with a ball to powder ratio of 14:1. The bulk samples were then divided individually into containers with each container consisting of a total specimen amounting to 4 g. The mixed powders were then compressed within a cylindrical die with a diameter of 20 mm using an Instron universal testing machine (INSTRON 3382, Shandong, China) at a compaction pressure of 302 MPa, which is equivalent to 95 kN. The compacted samples were eventually transferred for the sintering process with the assistance of the Carbolite Furnace. The major reduction in RHA was observed at the range of 560 to 600 °C following the Thermogravimetric Analysis (TGA) of RHA [16], whereas the TGA of the aluminum shows that the melting point of aluminum falls around 660 °C [17], thus the sintering temperature was set at 600 °C to ensure the removal of reinforcement material while avoiding the possibility of forming molten aluminum due to unintended melting. Therefore, the samples were, in the end, sintered at a temperature of 600 °C with a holding time of 20 min and a heating rate of 4 °C/min. Figure 2 shows the flow chart for the entire process of fabrication as well as the testing in this study.

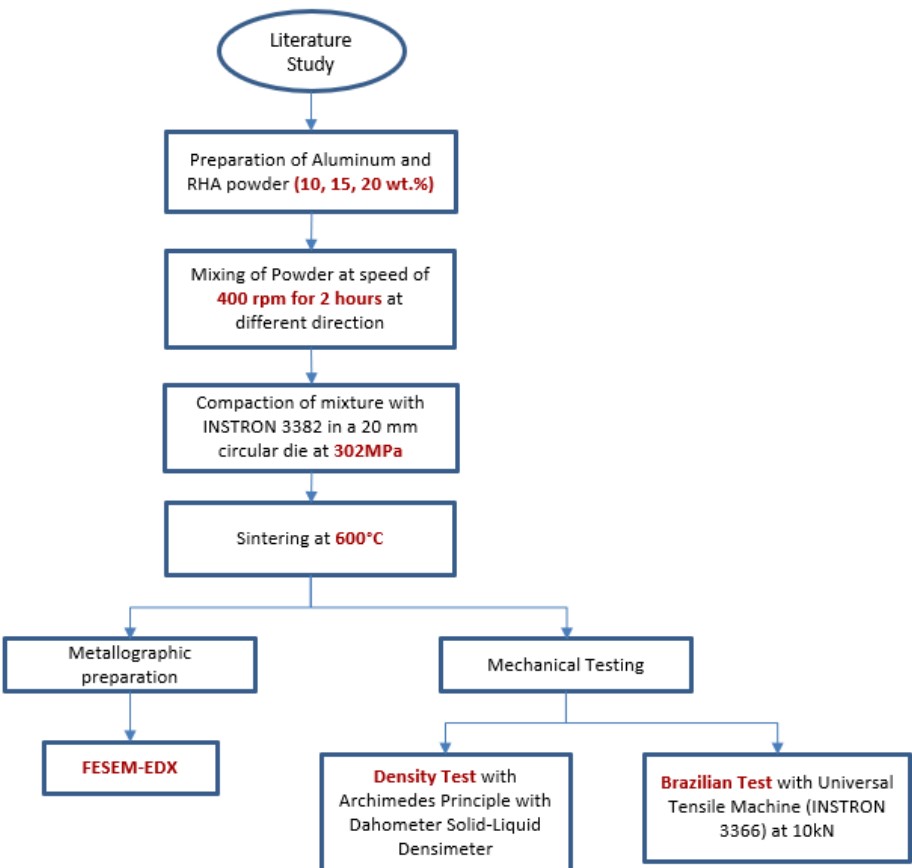

**Figure 2.** Complete flow chart of the study.

*2.2. Testing and Analysis*

The microstructure and the chemical composition of the samples from each RHA composition were then characterized using ultra-high image resolution with an FEI Nova Nano SEM 230 field emission scanning electron microscope (FESEM, SU8000, Hitachi, Tokyo, Japan) with energy-dispersive X-ray spectroscopy (EDX). Different magnifications were applied, including 3000×, 5000×, and 10,000×. The pore size obtained in each sample was evaluated by utilizing the microstructure image obtained from FESEM at a magnification of 10,000× with the aid of ImageJ software. The images were taken from the cross-section surface of the failed samples from the Brazilian Test.

As for tensile strength and porosity evaluation, the testing was carried out following ASTM D3967 and ASTM B962, respectively. The tensile strength of the samples was evaluated through Brazilian Testing with a specimen dimension of 20 mm in diameter and a height of 10 mm. Four different pellets were prepared for each RHA composition to obtain an average value of Tensile Strength and Orthogonal Stress applied vertically above the vertically positioned specimen. Diametric compression was carried out using an Instron universal testing machine (INSTRON 3366, Shandong, China)) with a maximum load capacity of 10 kN. In accordance with ASTM D3967, the crosshead rate should be set at a range from 0.1 mm/min to 1.0 mm/min [18]. Hence, through consideration, the crosshead rate was set at 0.5 mm/min in alignment with Brazilian Testing done by Ali et al. [19]. The individual Tensile Strength and Orthogonal Stress can then be evaluated through Equations (1) and (2) [18].

$$\sigma_y = -\frac{2P}{\pi dt} \tag{1}$$

$$\sigma_y = -3\sigma_x = \frac{6P}{\pi dt} \tag{2}$$

where $\sigma$ is in MPa indicating the tensile strength, $P$ is the applied force (N), $d$ is the diameter of the samples (mm), and $t$ is the thickness of the sample (mm).

For density and porosity, Archimedes' Principle was adapted with the help of the Dahometer Solid-Liquid Densimeter. The mass of the sample in atmospheric air was recorded as $m_1$, whereas the mass of the sample when submerged in water and the mass after removal from the water were recorded as $m_2$ and $m_3$, respectively [20]. The density, open porosity, and total porosity can then be computed according to Archimedes' Principle using Equations (3)–(5) [20].

$$\text{Density, } \rho \left( \frac{g}{cm^3} \right) = \frac{m_1}{m_3 - m_2} \times \rho_{H20} \tag{3}$$

$$\text{Open Porosity, } P_{\text{open}}(\%) = \frac{m_3 - m_1}{m_3 - m_2} \times 100 \tag{4}$$

$$\text{Total Porosity, } P_{\text{TOTAL}}(\%) = \left( 1 - \frac{\rho}{\text{Theoretical Density}} \right) \times 100 \tag{5}$$

## 3. Results and Discussion

### 3.1. Microstructure of Powder Mixture

The morphology was determined and identified through the assistance of FESEM-EDX; SU8000, Hitachi, Japan. It was noticed that the pore size generally increases when the RHA composition increases from 10 wt.% to 20 wt.%. The black voids were deduced to be the formed pores and it was noticed that the pore size increases from 516.15 nm to 861.58 nm. In comparison with the particle size of RHA, it was observed that the size increases by several folds, which can be inferred as the clumping of RHA when fabricating each sample. This was further explained by Mohanta et al. [12], where greater pore size typically implies that more connectivity was formed between two distinct or similar particles, thus leading to the formation of open pores with significant size. Nevertheless, the increasing trend of pore size was affirmed and accepted positively, as greater composition tends to encourage a greater amount of reinforcement material removal, thus nurturing a greater number of formed pores within the aluminum matrix. The obtained result aligns with Ali et al., wherein, when the RHA composition increased, larger elongated open pores were observed, which gave rise to the fabrication of an aluminum metal matrix composite with high porosity [19]. Figure 3 depicts the FESEM image of the cross-section of the obtained sample for each RHA composition under magnifications of 3000×, 5000×, and 10,000× in ascending order from left to right.

As for elemental composition, the element constituents for each powder mixture were found to be approximately similar, which denotes that no contaminants were identified from EDX equipment. In all three cases, large amounts of aluminum and oxygen were detected. Oxygen was found to be abundant within the aluminum matrix due to the presence of RHA, which consists majorly of silicon dioxide, thus contributing to the high oxygen content. Apart from that, aluminum is an extremely reactive metal that readily reacts with oxygen from atmospheric air, resulting in the formation of a much more stable compound known as aluminum oxide. Therefore, this leads to the ranking of elements in descending order from aluminum (Al) to oxygen (O) and silicon (Si) with weight percentages, as illustrated in Table 1. A certain percentage of carbon (C) was detected in the aluminum powder, as can be seen in the table. This is due to the penetration of the high-energy electron beam into the carbon tape which is used to attach the powder to the sample holder.

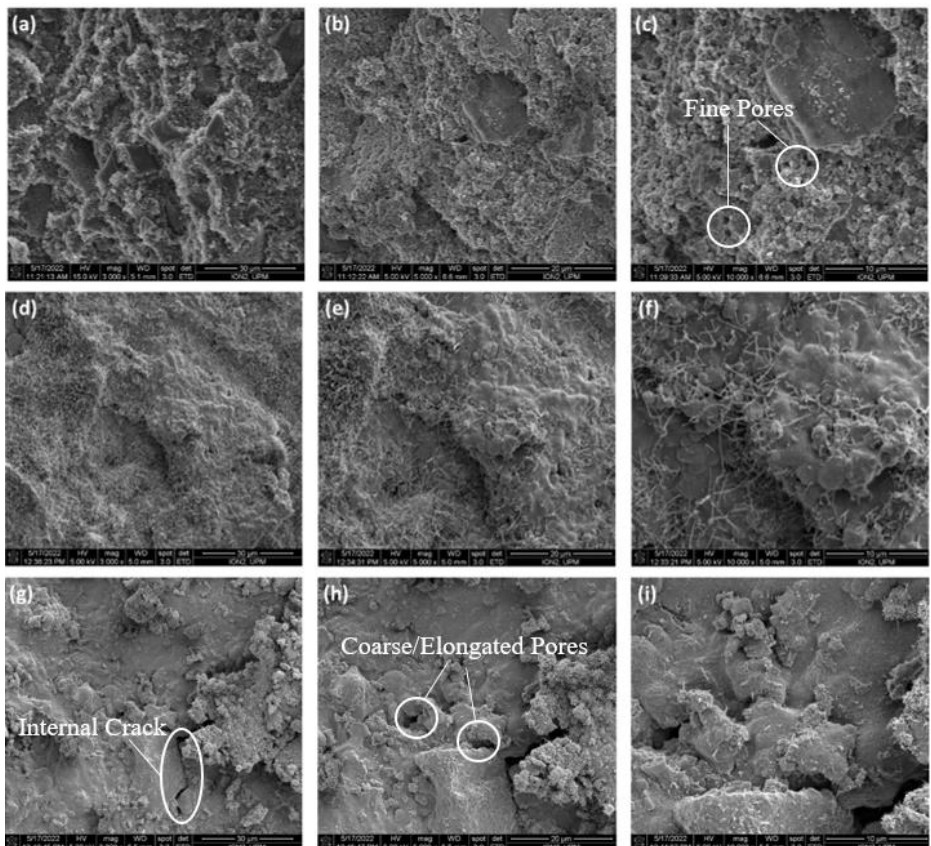

**Figure 3.** FESEM images of aluminum samples' cross-sections with magnifications of 3000×, 5000×, and 10,000× from left to right. (**a,d,g**) RHA-10 wt.%, (**b,e,h**) RHA-15 wt.%, (**c,f,i**) RHA-20 wt.%.

**Table 1.** Elements that were detected for each case through EDX analysis.

| Specimens | Qualitative Results, wt.% | | | | | | | | |
|---|---|---|---|---|---|---|---|---|---|
| | **Al** | **O** | **Si** | **Na** | **Mg** | **Cl** | **K** | **C** | **Ca** |
| Al powder | 58.32 | 11.63 | - | - | - | - | - | 30.03 | - |
| RHA | - | 57.22 | 40.52 | - | 0.62 | - | 0.59 | - | 1.05 |
| RHA-10 wt.% | 68.39 | 17.40 | 10.96 | - | - | - | - | - | - |
| RHA-15 wt.% | 44.04 | 51.19 | 1.05 | 0.61 | 1.13 | 0.45 | 1.53 | - | - |
| RHA-20 wt.% | 52.75 | 14.21 | 32.65 | - | - | - | 0.38 | - | - |

Nonetheless, the high amount of oxygen content was affirmed to be silicon dioxide following Shaikh et al., who also obtained a high amount of aluminum, silicon, and oxygen when aluminum and RHA were utilized in sample fabrication [8]. Apart from that, several minor elements such as nitrogen (N), magnesium (Mg), chlorine (Cl), potassium (K), and sodium (Na) can be detected in each case. Depending on the environment and process conditions during fabrication, different compounds might be formed along with the production phase. For instance, MAD-10 feldspars were detected most often in all three trials, which were implied as the formation of aluminosilicate due to the reaction between aluminum and silicon dioxide. Reaction between elements such as sodium, potassium, or calcium typically results in the formation of different feldspars where the common type of feldspars may include albite ($NaAlSi_3O_8$), sanidine ($KAlSi_3O_8$), orthoclase ($KAlSi_3O_8$), microcline ($KAlSi_3O_8$), and anorthite ($CaAl_2Si_2O_8$). Nevertheless, the motive of carbonaceous compound removal upon the sintering stage was achieved, as the carbon content was found to be minimal to none during testing.

*3.2. Mechanical Properties*

For theoretical density, the density increases for every 5 wt.% increase in RHA, since the theoretical density was identified to be 2.70 g/cm$^3$, 2.17 g/cm$^3$, and 1.026 g/cm$^3$ for aluminum, RHA, and magnesium stearate, respectively. The theoretical density for RHA-10 wt.%, RHA-15 wt.%, and RHA-20 wt.% were found to be 2.51 g/cm$^3$, 2.54 g/cm$^3$, and 2.56 g/cm$^3$ respectively. As for experimental density, the average density for each sintered pellet sample was recorded as 2.166 g/cm$^3$, 2.050 g/cm$^3$, and 2.169 g/cm$^3$ for samples with RHA composition increasing from 10 wt.% to 20 wt.% under Archimedes' Principle. Therefore, by utilizing Equation (5), the total porosity for every sample can be evaluated and acknowledged where the total porosity increases from 13.7% to 19.19% before decreasing to 15.38% when the RHA composition of 10 wt.% was enhanced to 20 wt.% at 5 wt.% intervals. It was then found that the trend of increasing total porosity with decreasing density aligned with the trend observed by several researchers, including Aida et al., who utilized magnesium as the target matrix and carbamide as the space-holding material [21]. The obtained trend was also identified to align with the initial hypothesis that inferred that when the composition of reinforcement material or pore-forming agent increases, the voids that are observed within the surface morphology of samples increase, which indicates the formation of aluminum metal matrix composites with higher porosity, thus lower density [12].

However, an interesting trend was observed when the RHA composition increased from 15 wt.% to 20 wt.%, where the total porosity of RHA-20 wt.% was observed to be reduced by approximately 4% with an increasing density of around 5 % when compared with RHA-10 wt.%. However, by observing the open porosity of each composition, the open porosity was recorded to be increasing at a steady rate from 4.52% to 4.64% and 6.95% for RHA compositions of 10 wt.%, 15 wt.%, and 20 wt.%, respectively. The obtained results indicate that although sintering may result in the densification of the specimen, thus leading to the elimination of holes within the sample matrix, a porous metal matrix typically consists of two different types of formed voids or channels which are known as open pores and closed pores, where the elimination of either one might lead to the deterioration of the total porosity of a produced or fabricated sample. Nevertheless, by visual inspection, a large crack was observed in Figure 3 for RHA-20 wt.%, which may also lead to obtaining results with high uncertainty and low accuracy, including the production of higher open porosity in comparison with RHA-10 wt.% and RHA-15 wt.%. The formed crack may be associated with the failure to consolidate the powder mixture during the powder compaction phase, which can be caused by several factors that result in the production of cracks and failures during the ejection phase [22]. To summarize, the trend of increasing porosity with decreasing density was observed as the RHA composition increased in each respective trial. Figure 4 illustrates the graph of experimental density and open porosity against the RHA composition of the fabricated aluminum metal matrix composite.

The tensile strength and orthogonal stress for each specimen can be evaluated using Equations (1) and (2). The average tensile strength for each composition was found to be 4.49 MPa, 5.19 MPa, and 5.10 MPa for RHA compositions of 10 wt.%, 15 wt.%, and 20 wt.%, respectively, with orthogonal stress increasing from 15.30 MPa to 15.56 MPa before decreasing to 13.46 MPa. The results were found to be aligned with the trend of obtained density and porosity, which affirms that increasing porosity typically affects the overall tensile strength of aluminum metal matrix composite due to the formation of a greater number of pores within the target matrix during the powder metallurgy phase. Apart from that, the dropping trend of mechanical strength upon reaching 15 wt.% was also obtained by Khamsuk et al. [23]. Regardless of other parameters, it is then assured that as the porosity increases or decreases, tensile strength tends to react similarly due to their directly proportional relationship. Apart from that, it was interesting to note that similar to density and porosity, RHA-20 wt.% tends to react differently in a negative trend, which can again be referred to as the result of several factors from errors during the ejection phase of the compaction process to the ununiform geometry of our involved samples. Nonetheless,

the trend of decreasing tensile strength with increasing RHA composition was determined along with the relationship between tensile strength, density, and porosity. Figure 5 shows the changes in average tensile strength for each sample with increasing RHA composition during Brazilian Testing.

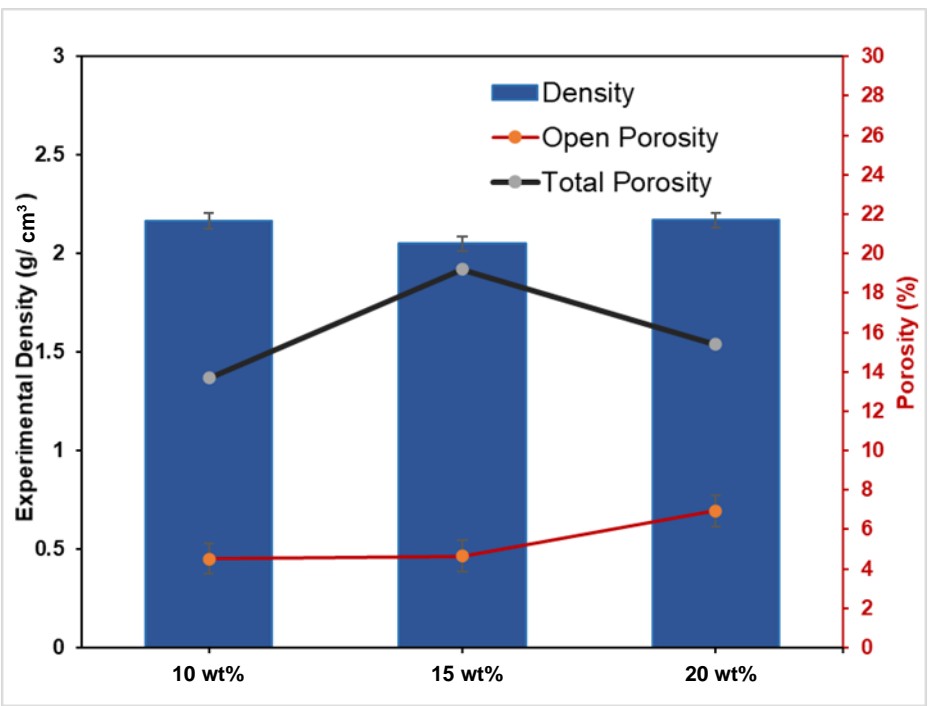

**Figure 4.** Graph of experimental density and open porosity against RHA composition.

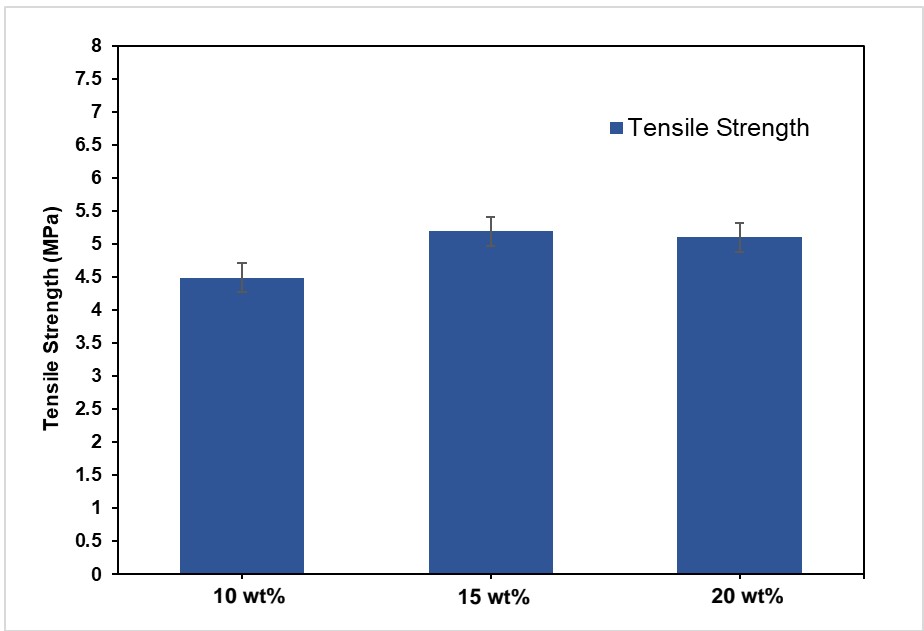

**Figure 5.** Graph of average tensile strength for each sample with increasing RHA composition during Brazilian Testing.

## 4. Conclusions

In conclusion, the main aim of the study, which was to fabricate porous aluminum with rice husk ash as a reinforcement material in different compositions, was achieved. Powder



metallurgy was utilized as the fabrication method in producing porous aluminum with different RHA compositions, which led to the production of porous aluminum with different densities, porosity, and mechanical strength. The tensile strength was recorded to increase from 4.49 MPa to 5.19 MPa before decreasing to 5.10 MPa for RHA compositions of 10 wt.%, 15 wt.%, and 20 wt.% respectively. On the other hand, the total porosity of the 15 wt.% rice husk ash aluminum sample was found to be the highest at 19.19 % in comparison with RHA-10 wt.% and RHA-20 wt.% with a total porosity of 13.7% and 15.38%, respectively. The morphology obtained from FESEM concurs that the presence of a significant number of open pores within RHA-20 wt.% porous aluminum with the identification of both coarse and elongated pores explains the result of high open porosity attained by RHA-20 wt.% specimens. Nevertheless, the results explain the inverse relationship between porosity and mechanical strength, whereby the proper control of parameters during fabrication will produce an aluminum metal matrix composite with ideal characteristics and properties. Further analysis might need to be performed to acknowledge the difference in strength as well as mechanical properties between a typical aluminum sample and an aluminum metal matrix composite fabricated with RHA to have a better understanding of the effect of the silicon dioxide within RHA on the overall functional improvement of the aluminum matrix.

**Author Contributions:** Conceptualization, O.J.L. and A.H.M.A.; methodology, A.H.M.A.; validation, O.J.L.; formal analysis, O.J.L.; investigation, O.J.L.; resources, A.H.M.A. and D.-W.J.; data curation, S.M.T. and M.H.S.; writing—original draft preparation, O.J.L.; writing—review and editing, A.H.M.A. and D.-W.J.; visualization, A.H.M.A.; supervision, A.H.M.A., S.M.T. and M.H.S.; project administration, A.H.M.A.; funding acquisition, A.H.M.A. and D.-W.J. All authors have read and agreed to the published version of the manuscript.

**Funding:** The authors would like to declare that this study and publication were supported by the Research Management Center of Universiti Putra Malaysia, Malaysia (UPM/GP-GPB/2017/9564200), (UPM/GP-IPB/2020/9688700), the Department of Mechanical and Manufacturing Engineering, UPM, and Jeju National University, Korea.

**Institutional Review Board Statement:** Not applicable.

**Informed Consent Statement:** Not applicable.

**Data Availability Statement:** Not applicable.

**Acknowledgments:** Research Management Center of Universiti Putra Malaysia, Department of Mechanical and Manufacturing Engineering, Faculty of Engineering, UPM, and Jeju National University, Korea.

**Conflicts of Interest:** The authors declare no conflict of interest.

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
