# Peer review of "Rice Husk Ash as Pore Former and Reinforcement on the Porosity, Microstructure, and Tensile Strength of Aluminum MMC Fabricated via the Powder Metallurgy Method"

_crystals, doi:10.3390/cryst12081100_

Round 1

Reviewer 1 Report

The SEM pictures of received RHA and Al powder before the experiments are missing; please add these figures to the paper.

The authors show the 10, 15, and 20 wt% samples. It would be nice to see the 0 wt% sample, I mean the sample without RHA. It would be good to know how the structure is modified when you change from 0 to 10 wt%.

Table 1. contains the EDS analysis of the mixture (? this is not written in the title.), but the „raw” materials EDX analysis is missing. We do not know where this high oxygen content is. Perhaps from the ash, but the Al powder also can contain oxygen.

Author Response

Item

Comment

Response/ Correction

1

The SEM pictures of received RHA and Al powder before the experiments are missing; please add these figures to the paper.

SEM image of the Al powder and ground RHA were included as Figure 1. Details on the size of the RHA have also been included in the Materials and Methods section.

“The obtained RHA was ground to smaller particle size using a mechanical blender or Na-tional MX-896TM dry blender to reduce the overall particle size before sieving with a 106-μm Retsch 200 x 50 mm 106 µm Test Sieves. which give rise to production of rice husk ash with an average particle size of 106 μm.”

2

The authors show the 10, 15, and 20 wt% samples. It would be nice to see the 0 wt% sample, I mean the sample without RHA. It would be good to know how the structure is modified when you change from 0 to 10 wt%.

Thank you for the suggestion. However, sample without RHA were not prepared in the process.

3

Table 1. contains the EDS analysis of the mixture (? this is not written in the title.), but the „raw” materials EDX analysis is missing. We do not know where this high oxygen content is. Perhaps from the ash, but the Al powder also can contain oxygen.

The raw material EDX data has been included in Table 1. Explanation has also been included for the raw materials.

Reviewer 2 Report

The authors presented an article about “Rice Husk Ash as Pore Former and Reinforcement on Porosity, Microstructure and Compression Strength of Aluminum MMC Fabricated via Powder Metallurgy Method ”. I think the paper is well organized and appropriate for “Crystals” journal but the paper will be ready for publication after major revision.

•          The abstract looks good. Please include all significance numerical results.

•          For the introduction, please add more current references and briefly explain them.

•          In the last paragraph of the introduction, it should be expressed the novelty of the study, the differences from the past in detail.

•          Results and discussion and conclusion parts are inadequate according to citation and analyze in detail. There should be the importance of the study in detail, comparison results with other approaches in literature, the success of the experimental and computational results.

•          Improve the results and discussion and conclusion parts.

•          Please fix the typographical and eventual language problems in paper.

•          The paper is well-organized yet there is a reference problem. First, your reference list contains no paper from “Crystals” journal. If your work is convenient for this journal’s context then there are many references from this journal. Secondly, cited sources should be primary ones. Namely, indexed area shows the power of a paper and directly your paper’s reliability. Please make regulations in this direction.

•          The article should be rearranged by taking into account the journal writing rules and citation rules.

*** Authors must consider them properly before submitting the revised manuscript. A point-by-point reply is required when the revised files are submitted.

Author Response

Item

Comment

Response/ Correction

1

The abstract looks good. Please include all significance numerical results.

Thank you for the comment. The abstract has been improved following the comments.

“The total porosity of 15 wt% was found to be the highest at 19.19% yet with the highest com-pressive strength at 5.19 MPa due to its low open porosity at 4.65%. In contrast, total porosity of 20 wt% was found to be the lowest slightly lower at 15.38% yet with the highest open porosity at 6.95% which gives rise to its low tensile strength at 5.10 MPa. Therefore, indicating that reducing open porosity through controlling the composition of reinforcement specifically tends to enhance the mechanical strength of aluminum metal matrix composites.”

2

For the introduction, please add more current references and briefly explain them.

Several new references have been added to improve the paper as mentioned.

3

In the last paragraph of the introduction, it should be expressed the novelty of the study, the differences from the past in detail.

Thank you. The statement has been included in introduction.

“Apart from that, though large number of studies can be found revolving around on fabrication of aluminum matrix composites with RHA, minimal research was found that generally focuses on fabrication of porous aluminum through powder metallurgy approach.”

4

Results and discussion and conclusion parts are inadequate according to citation and analyze in detail. There should be the importance of the study in detail, comparison results with other approaches in literature, the success of the experimental and computational results.

Thank you for the comment. Improvement has been made as highlighted in the draft. However, the experiment does not include computational simulation.

5

Improve the results and discussion and conclusion parts.

Thank you. The improvement has been highlighted in the draft submitted.

6

Please fix the typographical and eventual language problems in paper.

Thank you. The changes have been highlighted in the manuscript.

7

The paper is well-organized yet there is a reference problem. First, your reference list contains no paper from “Crystals” journal. If your work is convenient for this journal’s context then there are many references from this journal. Secondly, cited sources should be primary ones. Namely, indexed area shows the power of a paper and directly your paper’s reliability. Please make regulations in this direction.

Thank you for the comment. We have added few more references to improve the article and one reference from the journal Crystal. I am sorry that we could not find more papers related to this topic in the journal Crystal to add to. But we will keep that in mind for future publications.

8

The article should be rearranged by taking into account the journal writing rules and citation rules.

Thank you. Corrections has been made to address the mentioned mistake.

Reviewer 3 Report

It is required to expand the review part by adding sources dedicated specifically to the research part of the paper.

In the Materials and Methods section, it would be nice to describe in more detail the process of preparing and mixing powders. What size RHA powder was eventually used for mixing and sintering? What was its particle size distribution? Provide photographs of the original aluminum and RHA powders. What was the chemical composition of RHA?

Is photo 2 a photograph of the surface of a sintered sample? In this case, you should indicate this in the caption of the figure and the description in the text.

Table 1 shows data for RHA levels of 10%, 20%, and 30%, and all experimental work is done for 10%, 15%, and 20% RHA values. Why is there such a discrepancy?

In Table 1, your 20% RHA value is very different in composition from 10% and 30%, why such a difference? This also raises the question of the original chemical composition of RHA, what was? Table 1 on how many experimental samples were made?

Why did the article limit itself to only three RHA content values, such as 10%, 15%, and 20%?

This must also be indicated in the Materials and Methods section.

What were the sizes and shapes of the pores? And why didn't they take photos of the polished cut of the samples? In such photographs, it would be possible to see the pores, their shape, and size, as well as the distribution over the structure of the samples.

It is also necessary to improve the quality of Figure 2.

Author Response

Item

Comment

Response/ Correction

1

It is required to expand the review part by adding sources dedicated specifically to the research part of the paper.

Thank you. Improvement has been made to the draft as highlighted.

2

In the Materials and Methods section, it would be nice to describe in more detail the process of preparing and mixing powders. What size RHA powder was eventually used for mixing and sintering? What was its particle size distribution? Provide photographs of the original aluminum and RHA powders. What was the chemical composition of RHA?

Thank you for the comments. The FESEM images of the raw materials and the EDX data has been added in Figure 1 and Table 1. Details on the RHA after grinding has also been added in the Materials and Methods section.

“The obtained RHA was ground to smaller particle size using a mechanical blender or National MX-896TM dry blender to reduce the overall particle size before sieving with a 106 μm Retsch 200 x 50 mm 106 µm Test Sieves. which give rise to production of rice husk ash with an average particle size of 106 μm.”

3

Is photo 2 a photograph of the surface of a sintered sample? In this case, you should indicate this in the caption of the figure and the description in the text.

Thank you for the comment. The caption has been edited to:

“Figure 2 depicts the FESEM image for each RHA composition of the cross section of the obtained sample for each RHA composition under magnification of 3000x, 5000x and 10000x in ascending order from left to right.”

4

Table 1 shows data for RHA levels of 10%, 20%, and 30%, and all experimental work is done for 10%, 15%, and 20% RHA values. Why is there such a discrepancy?

There is a mistake in the original draft. This draft has been corrected. Table 1 replaced with 10 wt%, 15wt% and 20wt%.

5

In Table 1, your 20% RHA value is very different in composition from 10% and 30%, why such a difference? This also raises the question of the original chemical composition of RHA, what was? Table 1 on how many experimental samples were made?

The original RHA EDX data has been added to Table 1.

Around 8 pellets were prepared for each parameter. However only one pellet was tested from each parameter for EDX due to our constraint.

6

Why did the article limit itself to only three RHA content values, such as 10%, 15%, and 20%? This must also be indicated in the Materials and Methods section.

The three different samples were prepared with three different RHA compositions including RHA-10 wt% and Al-85 wt%, RHA-15 wt% and Al-80 wt% as well as RHA-20 wt% and Al-75 wt% as referred to Shaikh et al. (2019) [5] study.

7

What were the sizes and shapes of the pores? And why didn't they take photos of the polished cut of the samples? In such photographs, it would be possible to see the pores, their shape, and size, as well as the distribution over the structure of the samples.

Thank you for the comments. The author unfortunately did not prepare samples for polished cross section analysis. We would consider that for our future work.

8

It is also necessary to improve the quality of Figure 2.

Thank you. The quality of the image has been improved.

Round 2

Reviewer 1 Report

For the future, as I wrote, It would be nice to see the 0 wt% sample, I mean the sample without RHA. In this case, you have four points to identify your results. I recommend to the authors, in the future to prepare the 0 point of the experiments.

I can accept this paper in this form.

Author Response

Thank you for the comment Prof. We will take that in mind for our future works. Thank you Prof. 

Reviewer 2 Report

The authors made the desired corrections. In my opinion, the article can be published as it is.

Author Response

Thank you Prof. 

Reviewer 3 Report

It would be possible to add a couple more sources on the topic of the work in the Literature Review section.

In the methodology section, it is also necessary to make a reference that the chemical composition of RHA is given in Table 1 below.

Author Response

Thank you for the comments Prof. Action taken is as below:

1-It would be possible to add a couple more sources on the topic of the work in the Literature Review section.

Four new references have been added to the literature section. Changes has been highlighted in the submitted draft. 

2-In the methodology section, it is also necessary to make a reference that the chemical composition of RHA is given in Table 1 below.

Thank you Prof. The statement has been added to the methodology section, as highlighted in the draft submitted.